# Quality Evaluation of the 0.01° Multi-Source Fusion Precipitation Product and Its Application in Extreme Precipitation Event

**Zheng Wang [1], Yang Pan [1], Junxia Gu [1,\*], Yu Zhang [2] and Jianrong Wang [3]**

[1]  National Meteorological Information Center, Beijing 100081, China; wangz@cma.gov.cn (Z.W.); pany@cma.gov.cn (Y.P.)
[2]  Henan Meteorological Observation Data Center, Zhengzhou 450000, China; zhangyuc477201@cma.cn
[3]  Anhui Meteorological Information Center, Hefei 230000, China; faithwakin@sina.com
\*   Correspondence: gujx@cma.gov.cn

**Abstract:** High-resolution and high-quality precipitation data play an important role in Numerical Weather Prediction Model testing, mountain flood geological disaster monitoring, hydrological monitoring and prediction and have become an urgent need for the development of modern meteorological business. The 0.01° multi-source fusion precipitation product is the latest precipitation product developed by the National Meteorological Information Center to meet the above needs. Taking the hourly precipitation observation data of 2400 national automatic stations as the evaluation base, independent and non-independent test methods are used to evaluate the 0.01° multi-source fusion precipitation product in 2020. The product quality differences between the 0.01° precipitation product and the 0.05° precipitation product are compared, and their application in extreme precipitation events are analyzed. The results show that, in the independent test, the product quality of the 0.01° precipitation product and the 0.05° precipitation product are basically the same, which is better than that of each single input data source, and the product quality in winter and spring is slightly lower than that in summer, and both products have better quality in the east in China. The evaluation results of the 0.01° precipitation product in the non-independent test are far better than that of the 0.05° product. The root mean square error and the correlation coefficient of the 0.01° multi-source fusion precipitation product are 0.169 mm/h and 0.995, respectively. In the extreme precipitation case analysis, the 0.01° precipitation product, which is more consistent with the station observation values, effectively improves the problem that the extreme value of the 0.05° product is lower than that of station observation values and greatly improves the accuracy of the precipitation extreme value in the product. The 0.01° multi-source fusion precipitation product has better spatial continuity, a more detailed description of precipitation spatial distribution and a more accurate reflection of precipitation extreme values, which will better provide precipitation data support for refined meteorological services, major activity support, disaster prevention and reduction, etc.

**Keywords:** high-resolution and high-quality precipitation data; independent and non-independent test; the 0.01° multi-source fusion precipitation product; extreme precipitation event

## 1. Introduction

Precipitation data are the basis of weather and climate monitoring, climate change research, model prediction tests and meteorological and hydrological prediction. They play an extremely important role in flood season prediction, meteorological prediction, agricultural guidance and disaster prevention and reduction. With the rapid development of meteorological observation systems, more and more observation data and numerical model simulation data such as ground automatic weather station data, radar data and satellite data can be used, and various industries have higher and higher requirements for grid precipitation products. High-resolution and high-quality precipitation data have gradually

become an urgent need for the development of modern meteorological business [1]. Multi-source fusion precipitation products can combine the advantages of precipitation data from different sources, and have gradually become the mainstream trend in the development of high-quality precipitation products in the world in recent decades [2–11].

In recent years, many meteorological institutions in China have committed themselves to research the multi-source fusion technology, develop different kinds of 0.05° precipitation products, and significantly improve the quality of precipitation products in China. The National Meteorological Satellite Center of China has developed FY series satellite precipitation products based on the revision of the intelligent objective analysis method considering station distance and angle [12]. The National Meteorological Information Center of China, using the Probability Density Function matching (PDF) + Optimal Interpolation (OI) method of the U.S. Climate Prediction Center (CPC), has developed two-source fusion precipitation products based on ground station data and FY satellites data and two-source fusion precipitation products based on ground station data and CMORPH satellites data [12–19]. On the basis of two-source fusion precipitation products, using Probability Density Function matching (PDF) + Bayesian Multi-model Average (BMA) + Optimal Interpolation (OI) method, the National Meteorological Information Center has developed a series of three-source fusion precipitation products based on ground station data, radar data and satellite data [20,21]. As grid products with high precision, high quality and continuous time and space [1], multi-source fusion precipitation products have been widely used in the fields of model forecast testing [22,23], hydrological forecast [24] and meteorological live monitoring in provinces of China.

In order to further meet the needs of high-resolution and high-quality grid precipitation data in the fields of intelligent grid forecasting business development, refined meteorological services and disaster prevention and reduction, the National Meteorological Information Center has developed a 0.01° multi-source fusion precipitation product based on ground station data, radar data and satellite data by using Probability Density Function matching (PDF) + Bayesian Multi-model Average (BMA) + Spatial Downscaling (DS) + Optimal Interpolation (OI) method [15]. The product has further improved the quality and resolution of the fusion precipitation product, including adding more precipitation data sources, multi-source quality control of station observation data, quality control of fusion precipitation products, optimization of fusion parameters of multi-grid variational analysis, etc. Now the precipitation product has completed the business distribution to all provinces, and the data are connected to the China Integrated Meteorological Information Service System (CIMISS) data environment to provide services through the Meteorological Unified Service Interface Community (MUSIC) interface. In this study, to evaluate the quality and application effect of the product in China, independent and non-independent tests are used to evaluate the 0.01° multi-source fusion precipitation products in 2020, and the effect of the product on the characterization of extreme precipitation is analyzed.

## 2. Data and Processing

There are seven data sources of multi-source fusion precipitation products: (1) Observation data of ground automatic weather stations with hourly precipitation data of more than 60,000 automatic weather stations in China after quality control. (2) Satellite inversion precipitation products: The FY2 satellite inversion precipitation product developed by the National Satellite Meteorological Center, nominal projection, spatial resolution of about 4 km and time resolution of 1 h; The CMORPH satellite inversion precipitation product with global resolution of about 7 km and 30 min developed by the US Climate Prediction Center; The GsMAP satellite inversion precipitation product with global resolution of 1hour and 10 km developed by Japan Aerospace Exploration Agency (JAXA); The IMERG satellite inversion precipitation product with 30 min and 10 km resolution developed by NASA. (3) Radar precipitation estimation data: The national radar quantitative precipitation estimation product with hourly and 0.01° resolution developed by the Meteorological Observation Center of China Meteorological Administration; The national radar quantita-

tive precipitation estimation product with hourly and 0.01° resolution developed by the National Meteorological Information Center of China Meteorological Administration.

In view of the missing detection of erroneous precipitation data observed by automatic stations, after real-time quality control, the automatic weather stations hourly precipitation data also adopted the collaborative quality control technology of multi-source meteorological data, including consistency check between weather phenomena and precipitation and consistency check with radar estimated precipitation. In addition, the threshold lookup table of precipitation data and blacklist of ground automatic weather stations were established. The optimized quality control algorithm strengthens the screening of micro precipitation, the screening of false precipitation data and extreme outliers of products and improves the product quality in many aspects.

The optimal selection method is used in the fusion application of radar data and satellite data, and the weight coefficient of the linear fusion method that is adjusted by season and region is used in the Bayesian Multi-model Average of satellite data. The main development process of the 0.01° multi-source fusion precipitation product is shown in Figure 1.

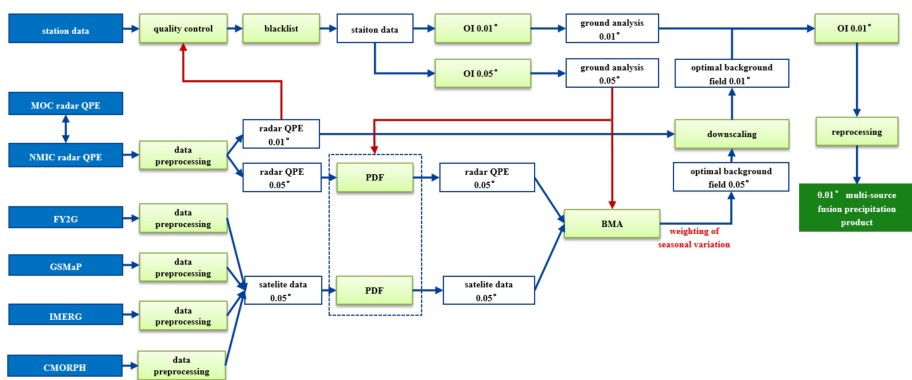

**Figure 1.** The main development process of the 0.01° multi-source fusion precipitation product.

The 0.01° and 0.05° resolution fusion analysis products can be downloaded through MUSIC, National Meteorological business intranet (http://idata.cma/) (accessed on 3 November 2021) and China Meteorological Data Network (http://data.cma.cn/) (accessed on 3 November 2021).

In addition, in the reprocessing of multi-source fusion precipitation products, some grid values will be replaced by the station precipitation value. In order to ensure the objectivity of the evaluation results, the evaluated product data are not replaced.

## 3. Product Evaluation Research

In this study, the hourly precipitation observation data of 2400 national automatic stations are used as the evaluation base, and the independent and non-independent test methods are used to evaluate the 0.01° multi-source fusion precipitation product in 2020. In the independent test, the observed precipitation data of 2400 national stations do not participate in the ground grid analysis and fusion and are used as the "truth" data of the test. In the non-independent test, the observed precipitation data of 2400 national stations participate in the ground grid analysis and fusion and are still used as the "truth" data of the test.

The statistical evaluation indexes in this paper are mean error (ME), root mean square error (RMSE) and Pearson correlation coefficient (R):

$$\mathrm{ME} = \frac{1}{\mathrm{n}} \sum_{i=1}^{n} (pi - gi) \tag{1}$$

$$\text{RMSE} = \sqrt{\frac{1}{n}\sum_{i=1}^{n}(pi - gi)^2} \tag{2}$$

$$R = \frac{\sum\limits_{i=1}^{n}(pi - \bar{p})(gi - \bar{g})}{\sqrt{\sum\limits_{i=1}^{n}(pi - \bar{p})^2}\sqrt{\sum\limits_{i=1}^{n}(gi - \bar{g})^2}} \tag{3}$$

In the formula, *gi* is the precipitation observation data of 2400 national stations which is regarded as the "true value" and *pi* is the precipitation value interpolated from each tested precipitation product to 2400 national stations.

### 3.1. Evaluation of Data from Different Sources and Time Series of Product Evaluation Results

The overall independent quality evaluation results of different resolution fusion precipitation products and the input data used in the products are shown in Table 1. The independent testing results in 2020 show that the quality of the 0.01° fusion precipitation product is basically the same as that of the 0.05° product, the Correlation Coefficients of both products are higher than 0.85, the Root Mean Square Error is less than 0.6 mm/h, and they are better than the quality of each single input datum. In the comparison of various input data, the quality of ground analysis is the best, followed by radar, and both are better than all kinds of satellite precipitation data. Among all kinds of satellite retrieved precipitation input data, the quality of the IMERG-L is the best, followed by the CMORPH, and both are better than other kinds of satellite precipitation products.

**Table 1.** The overall independent quality evaluation results of the fusion precipitation products and input data sources (2020).

| Number | Data | ME (mm/h) | | RMSE (mm/h) | | R | |
|---|---|---|---|---|---|---|---|
| | | 0.01° | 0.05° | 0.01° | 0.05° | 0.01° | 0.05° |
| 1 | Ground analysis | −0.013 | −0.015 | 0.576 | 0.597 | 0.857 | 0.845 |
| 2 | Radar QPE | −0.016 | −0.016 | 0.815 | 0.813 | 0.720 | 0.704 |
| 3 | FY2G | / | 0.017 | / | 1.478 | / | 0.245 |
| 4 | GSMaP-now | / | 0.018 | / | 1.288 | / | 0.244 |
| 5 | GSMaP-nrt | / | 0.004 | / | 1.284 | / | 0.299 |
| 6 | CMORPH | / | −0.034 | / | 1.102 | / | 0.366 |
| 7 | IMERG-L | / | 0.015 | / | 1.070 | / | 0.451 |
| 8 | Precipitation products | −0.010 | −0.007 | 0.524 | 0.519 | 0.851 | 0.854 |

Figure 2 shows the evaluation result time series of independent tests of fusion precipitation products from January to December 2020. In the whole year, there are little differences in the quality of the 0.01° fusion precipitation product and the 0.05° fusion precipitation product. In winter and spring, due to the suspension of the Tipper rain gauge in northern China (about 60,000 stations across the country are reduced to about 40,000 stations, as shown in Figure 3), and the insufficient inversion and estimation ability of satellite and radar detection means for solid precipitation, the quality of the products is affected, and the correlation coefficient is unstable. The root mean square errors of the 0.01° product and the 0.05° product are relatively consistent, and because the root mean square error is closely related to precipitation, the root mean square error in summer is higher than in winter and spring. Compared with the 0.05° precipitation product, the mean error of the 0.01° precipitation product in summer is slightly lower, which is closely related to the analysis radius of ground analysis of the two resolution products.

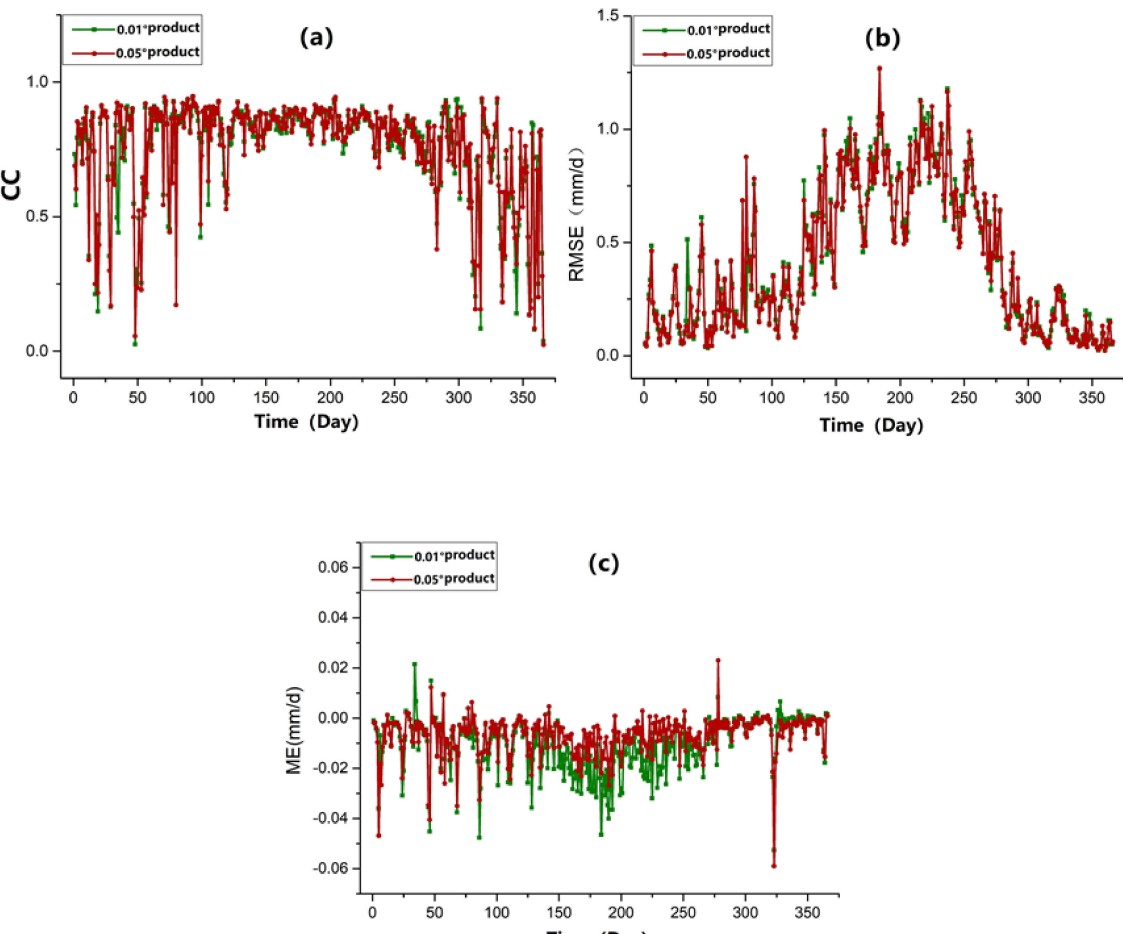

**Figure 2.** The evaluation results time series of independent test of fusion precipitation products in 2020 ((**a**): correlation coefficient, (**b**): root mean square error, (**c**): mean error, green line represents the 0.01° precipitation product, green line represents the 0.05° precipitation product).

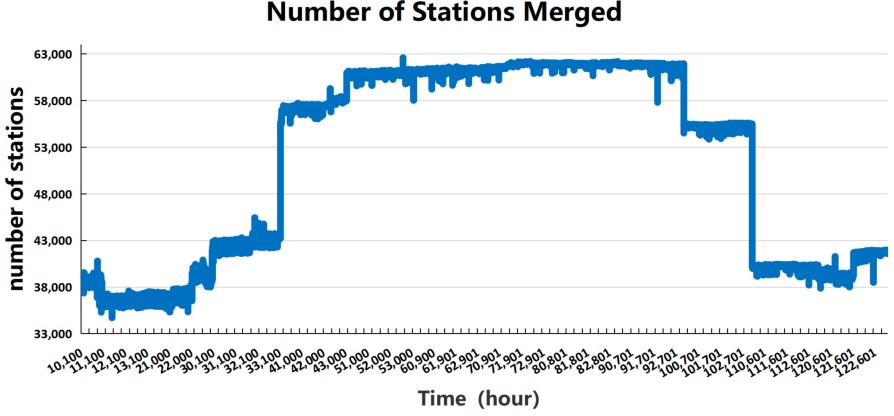

**Figure 3.** Number of automatic stations merged in the 0.01° precipitation product every hour in 2020. (10,100 represents UTC 00:00 on 1 January, and 122,601 represents UTC 01:00, 26 December).

*3.2. Spatial Distribution of Product Evaluation Index*

The spatial distribution of independent test evaluation results of fusion precipitation products in 2020 is shown in Figure 4. The spatial distribution of independent test evaluation results of the 0.01° fusion precipitation product and the 0.05° fusion precipitation

product is basically the same. In terms of the correlation coefficient (a1 and b1), the correlation coefficient in the dense area of stations in the east can basically exceed 0.8, and the correlation coefficient in the north and west is relatively low, which is related to the sparse distribution of ground stations and relatively few precipitation data in this area. The spatial distribution of the root mean square error is affected by precipitation. The root mean square error in the southeast is greater than that in the west and north with less precipitation (a2 and b2). The proportion of stations with root mean square error of 0~0.8 mm/h for 0.01° and 0.05° products is 90.41% and 89.56%, respectively. In the spatial distribution of mean error (a3 and b3), stations with absolute deviation greater than 0.05 mm/h are mainly distributed in areas with large annual precipitation in 2020. The proportion of stations with mean error of the 0.01° product and the 0.05° product between −0.05~0.05 mm/h are 96.08% and 95.15%, respectively. In addition, the 0.05° product has more positive mean error values. The proportion of stations with mean error of the 0.01° product and the 0.05° product greater than 0.05 mm/h is 0.7% and 2.0%, respectively, which is consistent with the small overall mean error of the 0.01° product in Figure 2 and Table 1. As can be seen from the spatial distribution of independent test results (Figure 4), the quality of the 0.01° product is slightly better than that of the 0.05° product, and both products have better quality in the east.

### 3.3. Comparison of Independent and Non-Independent Tests

Taking July 2020 as an example, the independent test and non-independent test results of the 0.01° fusion precipitation product and the 0.05° fusion precipitation product are analyzed and compared, as shown in Table 2. Test results in July 2020 show that independent test results of the 0.01° multi-source fusion precipitation product are basically consistent with those of the 0.05° product, and the evaluation results of the non-independent test indicate that the 0.01° multi-source fusion precipitation product is obviously better than the 0.05° product. In the non-independent test, the product quality of the 0.01° product and the 0.05° product was improved, and the product quality of the 0.01° product was improved more significantly.

**Table 2.** Comparison of independent test and non-independent test results of fusion precipitation products (July 2020).

| Number | Data | ME (mm/h) | RMSE (mm/h) | R |
| --- | --- | --- | --- | --- |
| 1 | 0.01° (non-independent) | 0.0001 | 0.169 | 0.995 |
| 2 | 0.05° (non-independent) | −0.0091 | 0.681 | 0.907 |
| 3 | 0.01° (independent) | −0.0253 | 0.825 | 0.863 |
| 4 | 0.05° (independent) | −0.0134 | 0.799 | 0.869 |

Correlation coefficient, root mean square error and mean error time series of the independent test and the non-independent test of the 0.01° fusion precipitation product and the 0.05° fusion precipitation product in July 2020 are shown in Figure 5. It can be seen from the time series, the independent test results of the 0.01° multi-source fusion precipitation product are basically consistent with those of the 0.05° product, and non-independent test results of the 0.01° product are slightly worse than those of the 0.05° product. In the non-independent test, the 0.01° precipitation product is much better than the 0.05° product, which means the precipitation of the 0.01° product is more consistent with the station observation values. For example, the root mean square error and the correlation coefficient of the non-independent test of the 0.01° multi-source fusion precipitation product are 0.169 mm/h and 0.995, respectively, and the root mean square error and the correlation coefficient of the non-independent test of the 0.05° product are 0.681 mm/h and 0.907, respectively. In addition, compared with other evaluation results, non-independent test results of the 0.01° product are much more stable.

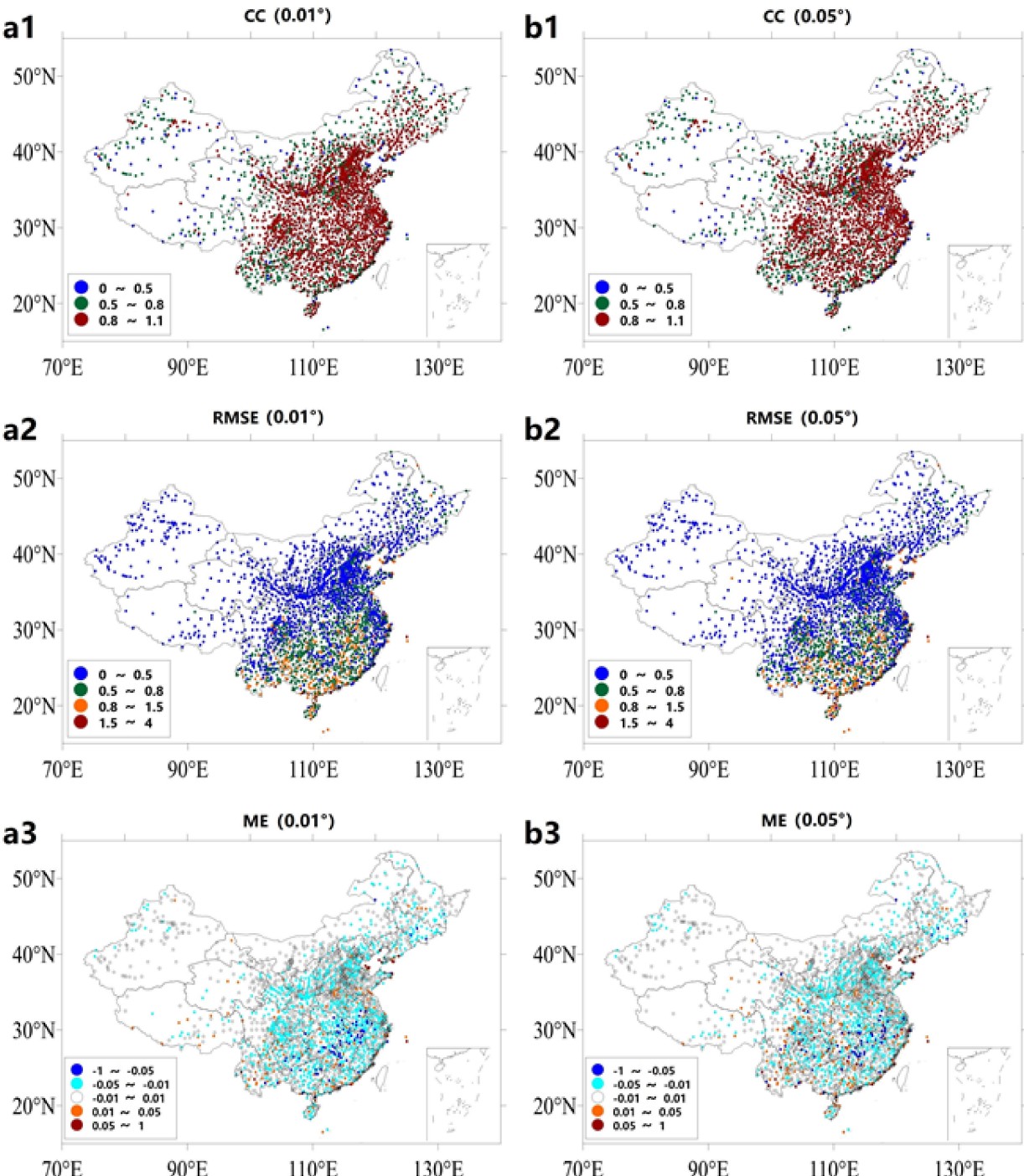

**Figure 4.** The spatial distribution of independent test evaluation results of fusion precipitation products in 2020. ((**a1**,**b1**): correlation coefficient, (**a2**,**b2**): root mean square error, (**a3**,**b3**): mean error).

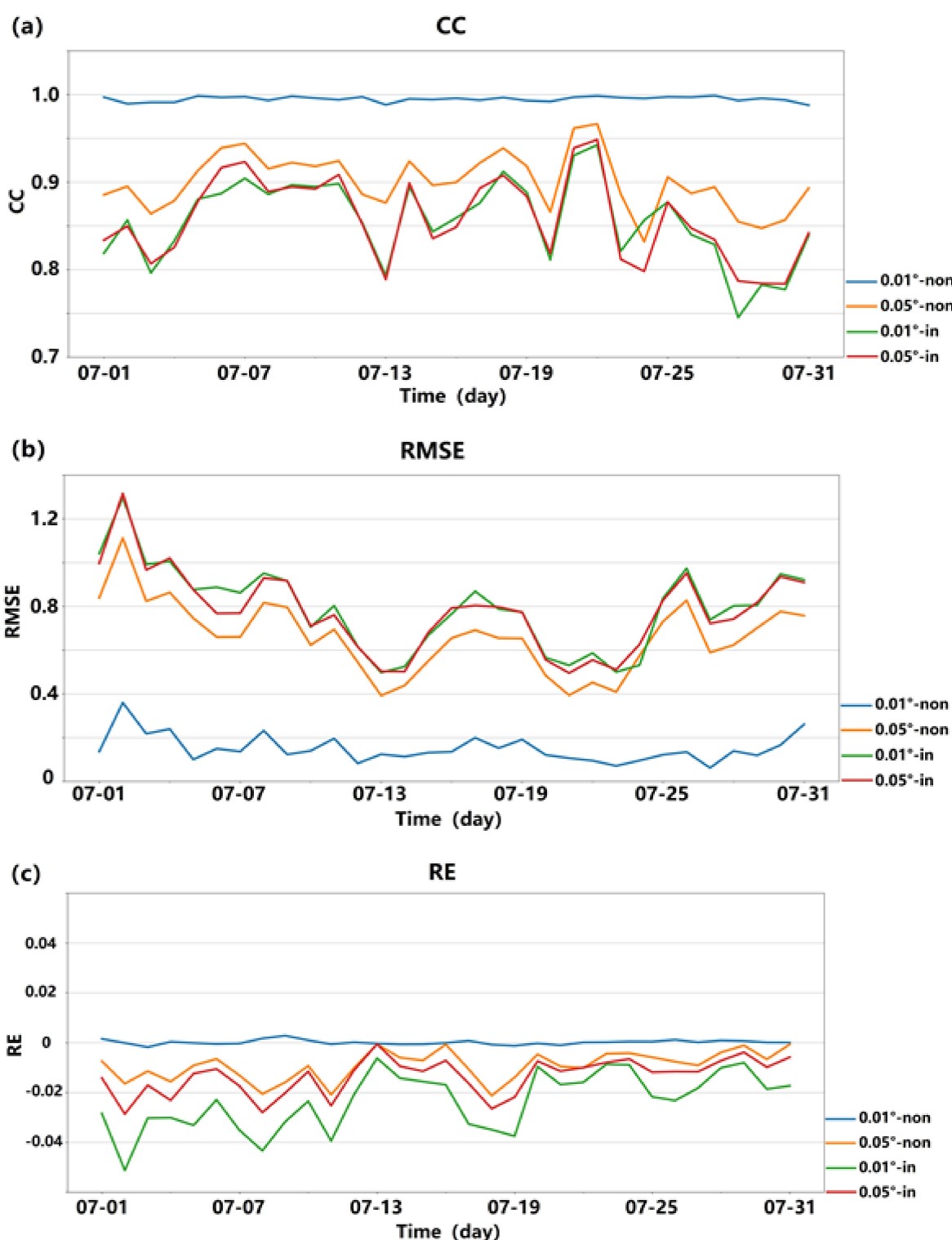

**Figure 5.** Time series of independent and non-independent test result of fusion precipitation products ((**a**): correlation coefficient, (**b**): root mean square error, (**c**): mean error, blue line: 0.01° non-independent, orange line: 0.05° non-independent, green line: 0.01° independent, red line: 0.05° independent).

## 4. Application of Fusion Precipitation Products in Extreme Precipitation Event

Taking the 7.20 rainstorm event in Henan in 2021 as an example, the application effect of the 0.01° precipitation product and the 0.05° precipitation product in this event is

analyzed. Figure 6 shows the spatial distribution of the 0.01° precipitation product and the 0.05° precipitation product in Henan on 20 July 2021. It can be seen from the figure that the spatial distribution of the 0.01° precipitation product and the 0.05° precipitation product is similar on the whole, and both have better spatial continuity than that of station data. The 0.01° multi-source precipitation product describes the spatial distribution of precipitation more finely and reflects the extreme precipitation more continuously and accurately. In the figure, on 20 July, the 24 h cumulative extreme precipitation of the station and the 0.01° precipitation product was 687.9 mm, which of the 0.05° precipitation product was only 580.2 mm. The spatial location of the 0.01° product extreme precipitation is more matched with that of station extreme precipitation. It can be seen that the 0.01° multi-source fusion precipitation product effectively improves the problem that the extreme precipitation of the 0.05° product is smaller, and greatly improves the accuracy of extreme precipitation of fusion precipitation products.

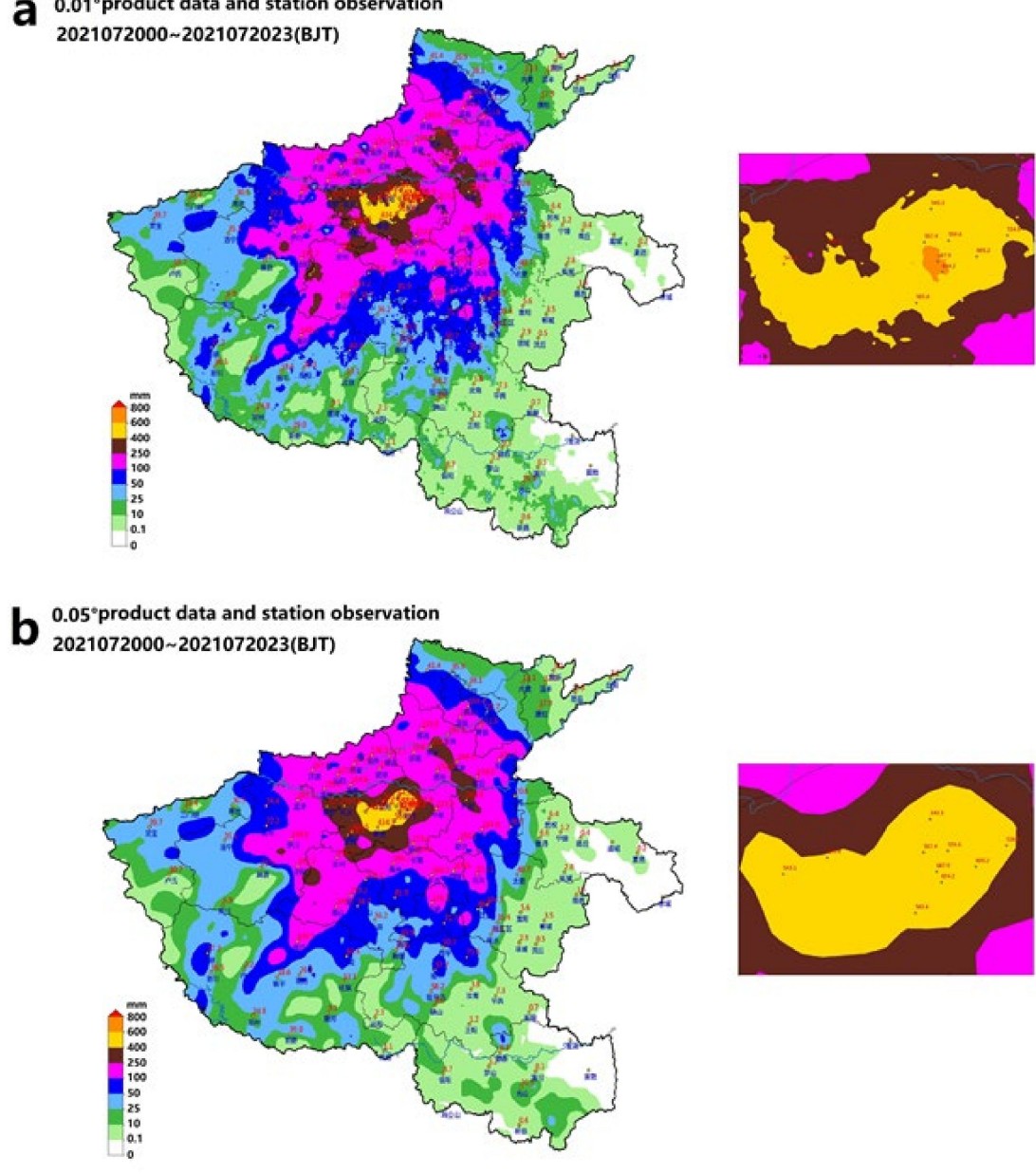

**Figure 6.** The spatial distribution of 0.01° and 0.05° precipitation products in Henan on 20 July 2021 (the picture on the right is an enlarged picture of the extreme precipitation region).

## 5. Conclusions

On the basis of the 0.05° three-source fusion precipitation product based on ground station data, radar data and FY2G satellite data, the National Meteorological Information Center has developed the 0.01° multi-source fusion precipitation product by using Probability Density Function matching (PDF) + Bayesian Multi-model Average (BMA) + Spatial Downscaling (DS) + Optimal Interpolation (OI). Taking the hourly precipitation observation data of 2400 national automatic stations as the evaluation base, independent and non-independent test methods are used to evaluate the 0.01° multi-source fusion precipitation product in 2020, the differences between the 0.01° multi-source fusion precipitation product and the 0.05° three-source fusion precipitation product are compared, and the spatial fineness and extreme value accuracy of extreme precipitation portrayed by precipitation products are analyzed. The main conclusions are as follows:

(1) From the overall independent test results in 2020, the quality of the 0.01° fusion precipitation product is basically the same as that of the 0.05° product. Both products are better than that of each single input data source. Among all data sources, the ground analysis quality is the best, followed by radar data, and the IMERG satellite precipitation data is the best among all satellite data sources. Both products have better quality in summer than that in winter and spring, and better quality in the east in China than that in the west.

(2) Independent test results of the 0.01° fusion precipitation product are basically consistent with those of the 0.05° product, which are slightly worse than the non-independent test results of the 0.05° product. The evaluation results of the 0.01° fusion precipitation product in the non-independent test are far better than those of the 0.05° product, which means the precipitation of 0.01° product is more consistent with the station observation values.

(3) The 0.01° multi-source fusion precipitation product has better spatial continuity, more detailed description of precipitation spatial distribution and more accurate embodiment of precipitation extreme value, which effectively improves the problem of the small extreme value of the 0.05° product and greatly improves the accuracy of precipitation extreme value.

High-quality and high-timeliness of the 0.01° multi-source fusion precipitation product in China will provide real-time precipitation data support for the upgrading of China's intelligent grid forecasting business to 0.01° resolution, and will play a significant economic and social benefit in disaster prevention and mitigation such as flood control and drought relief, refined meteorological services and guarantee of major activities. At present, the 0.01° multi-source fusion precipitation real-time product has been commercialized and is available for download. Although the effect of the 0.01° multi-source precipitation product in extreme precipitation events is preliminarily analyzed in this paper, there is still a lack of more detailed research, which can further analyze the applicability of the 0.01°multi-source precipitation product in extreme precipitation in different seasons and different precipitation levels. These contents will be reflected in future research, and this research is already in progress.

**Author Contributions:** J.G., Y.P. and Z.W. designed the study; Z.W. and J.W. performed the precipitation data preparation and analysis; Y.Z., J.W. and Z.W. draw the figures and pictures; the manuscript was written by Z.W. with significant contributions from all coauthors; the manuscript was revised by Z.W., J.G. and Y.P. All authors have read and agreed to the published version of the manuscript.

**Funding:** This research was funded by the National Key Research and Development Project, grant number 2018YFC1506604, and the Special Program for Innovation and Development of China Meteorological Administration, grant number CXFZ2021Z007.

**Institutional Review Board Statement:** Not applicable.

**Informed Consent Statement:** Not applicable.

**Data Availability Statement:** Not applicable.

**Acknowledgments:** This work was supported by the National Key Research and Development Project (2018YFC1506604) and the Special Program for Innovation and Development of China Meteorological Administration (CXFZ2021Z007).

**Conflicts of Interest:** The authors declare no conflict of interest.

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
