# Peer review of "Quality Evaluation of the 0.01° Multi-Source Fusion Precipitation Product and Its Application in Extreme Precipitation Event"

_sustainability, doi:10.3390/su14020616_

Round 1
Reviewer 1 Report
Summary
In this paper, fusion precipitation products based on ground-based measurements from automatic weather stations, precipitation radar data and satellite data at resolutions of 0.01° and 0.05° have been compared. The correlation coefficient, mean error and root mean squared error were calculated to compare the fusion precipitation product data with the measurements of precipitation firstly by not including the ground-based precipitation measurements in the fusion product, and secondly by including the ground-based precipitation measurements in the fusion product. It is found that the precipitation product compares better with the actual measurements when the measurements are included in the fusion product, particularly at the finer resolution (0.01°).
This study is a clear and useful assessment of a precipitation product that is widely used and has many applications.
Major Issues
There are no major issues to be addressed.
Minor Issues
Please address the following:
- Abstract: Please include a sentence explaining what the precipitation product is.
- Throughout the manuscript, please do not use random capital letters, e.g. line 36: Hydrological Prediction. Also, where a semi colon is used before a capital letter, the semi-colon should be changed to a full stop.
- The two precipitation products being compared are identified as the 0.01 and 0.05 degree products in the abstract. Please can these identifying names be used throughout the manuscript and in particular in the descriptions in lines 49-85.
- Line 87 onwards: Please can the numbering of the data sources correlate to the six mentioned.
- Line 132: The type of correlation coefficient calculated needs to be stated. Was it confirmed that the data is parametric/non-parametric?
- Figure 1: The abbreviations used for the axis titles should be written in full in the caption.
- Figure 2: The time period shown in this figure needs to be clarified. It looks like data is shown for 112 500 hours (almost 13 years). Which years are these? The text (line 152 on) implies that a seasonal variation is shown, which does not correspond with the 13 years.
- It is not clear why the independent tests are calculated for a year, while the non-independent tests are only calculated for a month. Please explain.
- Figure 5: The quality of this figures needs to be improved so that the station measurements are legible. Also, explanations need to be given in the caption to aid in the interpretation of the figure e.g. the total daily rainfall recorded by the automatic weather stations is shown in red.
- This manuscript needs to be thoroughly edited to correct grammar and punctuation errors.
Author Response
Dear editor, the response to the reviewer's comments is in the attachment.

Reviewer 2 Report
December 07, 2021
Manuscript: ‘Quality Evaluation of the 0.01° Multi-Source Fusion Precipitation Product and its Application in Extreme Precipitation Event’
In this paper, a cross-validation of multi-source precipitation products against ground observation data was applied over CHINA in 2020. Unlike other similar articles, this work applied independent and non-independent test methods at hourly time scale. I think that this is a relevant topic lies within the scope of the MDPI . The article is very well organized and neatly written with the appropriate scientific content.
********************************
Title: it fits perfectly the paper content.
Abstract: it is quite adjusted to the paper content.
Line 21: for clarity, indicate ‘in the East in China’.
Lines 21-26: for clarity, add at least one or two performance metrics to support this statement. For example, the root mean square error and correlation coefficient.
Introduction: it provides sufficient background and includes relevant references on satellite-based precipitation estimates, highlighting their limitations and strengths and the need to assess the multi-source fusion precipitation products against the ground-based precipitation data. Objectives and the novelty are clearly presented.
Line 80: indicate the meaning of ‘CIMISS´.
Line 81: indicate the meaning of ‘MUSIC´.
Lines 83-85: Some previous studies have compared the performance of multi-source precipitation estimates against ground-based precipitation data at monthly time scale. This was not, however, the case in this study. Why?
Data and Processing: the study area, datasets and methods have been clearly described. For clarity, I think that this section could be significantly improved if the authors add a flow chart with the different methods described in text, highlighting inputs, applied techniques, and outputs so that readers could understand this section easier.
Line 130: fix ‘…data are not replaced by the observed values of stations.’
Line 135: for this type of works, it is strongly advised to examine the rain-detection capability through categorical metrics (e.g., Probability of detection POD; false alarm ratio FAR; critical success index CSI; and bias score BS)
Product Evaluation Research:
Line 130: fix ‘data are not replaced by the observed values of stations.’
Line 182: fix ‘…between -0.05 and 0.05mm/h is…’
Line 217: what is the statistical meaning of ‘more stable’? Explain briefly.
Conclusion: these are clearly presented with an adequate narrative.
Author Response

(The authors gave the same response as above.)

Round 2
Reviewer 2 Report
December 28, 2021
The authors have revised the manuscript based on most of my comments. Therefore, I support this revised version for publication in MDPI sustainability.
